

# Genetic characterization of the prion protein gene in camels (*Camelus*) with comments on the evolutionary history of prion disease in Cetartiodactyla

Emily A. Wright[1], Madison B. Reddock[2], Emma K. Roberts[2,3], Yoseph W. Legesse[4,5], Gad Perry[6] and Robert D. Bradley[1,2]

[1] Natural Science Research Laboratory, Museum of Texas Tech University, Lubbock, TX, United States of America
[2] Department of Biological Sciences, Texas Tech University, Lubbock, TX, United States of America
[3] Climate Center, Texas Tech University, Lubbock, TX, United States of America
[4] School of Animal and Range Sciences, Haramaya University, Dire Dawa, Ethiopia
[5] Institute of Pastoral and Agropastoral Development Studies, Jigjiga University, Jigjiga, Ethiopia
[6] Department of Natural Resources Management, Texas Tech University, Lubbock, TX, United States of America

Corresponding author
Emily A. Wright,
emily.a.wright@ttu.edu

## ABSTRACT

Transmissible spongiform encephalopathies (TSEs) are a fatal neurogenerative disease that include Creutzfeldt–Jakob disease in humans, scrapie in sheep and goats, bovine spongiform encephalopathy (BSE), and several others as well as the recently described camel prion disease (CPD). CPD originally was documented in 3.1% of camels examined during an antemortem slaughterhouse inspection in the Ouargla region of Algeria. Of three individuals confirmed for CPD, two were sequenced for the exon 3 of the prion protein gene (PRNP) and were identical to sequences previously reported for *Camelus dromedarius*. Given that other TSEs, such as BSE, are known to be capable of cross–species transmission and that there is household consumption of meat and milk from *Camelus*, regulations to ensure camel and human health should be a One Health priority in exporting countries. Although the interspecies transmissibility of CPD currently is unknown, genotypic characterization of *Camelus PRNP* may be used for predictability of predisposition and potential susceptibility to CPD. Herein, eight breeds of dromedary camels from a previous genetic (mitochondrial DNA and microsatellites) and morphological study were genotyped for *PRNP* and compared to genotypes from CPD–positive Algerian camels. Sequence data from *PRNP* indicated that Ethiopian camels possessed 100% sequence identity to CPD–positive camels from Algeria. In addition, the camel *PRNP* genotype is unique compared to other members of the Orders Cetartiodactyla and Perissodactyla and provides an in–depth phylogenetic analysis of families within Cetartiodactyla and Perissodactyla that was used to infer the evolutionary history of the *PRNP* gene.

## INTRODUCTION

Spongiform encephalopathies are a fatal neurogenerative disease (*Prusiner, 1982*; *Prusiner, 1998*) that include Creutzfeldt–Jakob disease and Kuru in humans, scrapie in domestic sheep and goats, chronic wasting disease (CWD) in cervids, bovine spongiform encephalopathy (BSE), transmissible mink encephalopathy, feline spongiform encephalopathy, among others (*Abdalla & Sharif, 2022*; *Aguzzi & Polymenidou, 2004*; *Collinge & Clarke, 2007*; *Davenport et al., 2015*; *Greenlee & Greenlee, 2015*). Spongiform encephalopathies can be contracted through a variety of means: (1) consumption of infected flesh or contact with bodily fluids (transmissible, *Collins, Lawson & Masters, 2004*; *Haywood, 1997*; *Weissmann, 1999*), (2) genetic transfer of a mutated prion gene from one or both parents to offspring (familial, *Nitrini et al., 1997*; *Riek et al., 1998*), or (3) spontaneous production of an alternative prion protein (sporadic, *Brown et al., 2006*; *Casalone et al., 2004*). Additionally, dietary intake may influence transmission of prion diseases through consumption of infected animal products (meat, milk, *etc.*) or through infectious prions on or within plants and other biotic and abiotic material in the environment (*Bartelt-Hunt, Bartz & Yuan, 2023*; *Gough & Maddison, 2010*; *Inzalaco et al., 2023*; *Johnson et al., 2011*; *Konold et al., 2008*; *Kuznetsova et al., 2023*; *Lacroux et al., 2008*; *Prusiner, 1997*).

Evidence, obtained from the genotypic characterization of the exon 3 region of the prion protein gene (*PRNP*), has been relevant in determining the distribution of populations susceptible to TSE infection and in managing the spread of prion diseases (*Arifin et al., 2023*; *Buchholz et al., 2021*; *Fernandez-Borges, Erana & Castilla, 2018*; *Goldmann, 2008*; *Jewell et al., 2005*; *Mead et al., 2009*; *Otero et al., 2021*; *Perucchini et al., 2008*). The most common isoform, PrP$^c$, is inherited and is present during embryogenesis (*Westergard, Christensen & Harris, 2007*). However, mutated, protease–resistant isoforms (PrP$^{Sc}$) cause abnormal folding of the prion protein, aggregations of amyloid plaques (*Horwich & Weissman, 1997*), and ultimately the fatal presentation of a prion disease. Although the function of PrP remains unknown, the protein is involved with the circadian rhythm, homeostasis of metal ions, mitochondria, and myelin, intercellular signaling, and neuroprotection (reviewed in *Kovač & Šerbec, 2022*).

Some mammalian species have amino acid substitutions that may confer low susceptibility in wild populations. For example, there is evidence of strong salt bridges that link the $\beta$2- $\alpha$2 loop of the prion protein to suggest that water buffalo (*Bubalus bubalis*) has low susceptibility to TSEs similar to members of Canidae, Equidae, Leporidae, Mustelidae, and Suidae (*Zhang, Wang & Chatterjee, 2016*). However, most members of Suborder Ruminantia are thought to be highly susceptible to prion diseases; codon positions A136V, R154H, and R171Q/K as well as Q95H, S96G, and S225F are known to be important in the susceptibility of domestic sheep and North American deer, respectively (*Belt et al., 1995*; *Goldmann, 2008*; *Jewell et al., 2005*). Given the recent increase in CWD cases in the US and other prion diseases in Old-World ruminants, this is a critical area for determining species that might be increasingly at risk for prion exposure.

According to *Köhler-Rollefson (1991)*, Dromedary camels (*Camelus dromedarius*) have been extinct in the wild for approximately 2,000 years and have been under considerable
exploitation by humans. The population structure of and subsequent underlying genetic and evolutionary forces on Dromedary camels most likely has been human mediated for millennia (*Köhler, 1981*). In Ethiopia, populations of Dromedary camels (*Camelus dromedarius*) are mostly restricted to the Ethiopian regional states of Afar, Oromia, and Somali (*Abebe, 2001*). Although Dromedary camels are, in some instances, free-ranging, these populations and those under captive operations are actively maintained and used for pastoralism, including the production of milk and the sales for pack animals or slaughter (*Habte et al., 2021*; *Kena, 2022*; *Mirkena et al., 2018*).

In 2018, a novel camel prion disease (designated by *Babelhadj et al. (2018)* as CPD, termed CPrD by *Khalafalla (2021)*) in Dromedary camels was detected in the Ouargla abattoir (slaughterhouse) in Algeria, using traditional histological, immunohistochemical, and western blot techniques (*Babelhadj et al., 2018*). DNA sequences obtained from the *PRNP* gene were examined and then used to generate a genotype of CPD positive individuals; however, the authors made no inference from those data as unfortunately no CPD-negative individuals were sequenced for the *PRNP* gene (*Babelhadj et al., 2018*). Based on this initial study, *Babelhadj et al. (2018)* and *Watson et al. (2021)* suggested several hypotheses (*e.g.*, CPD naturally developed and was not related to scrapie or BSE, prion-contaminated waste dumps as a source of food in the Ouargla region, *etc.*) to explain the occurrence of CPD in Algeria; however, no patterns for transmission pathways were identified (*Orge et al., 2021*).

With the confirmed case of prion disease in Dromedary camels in Algeria (*Babelhadj et al., 2018*; *Khalafalla, 2021*) and a second case reported in Tunisia (*World Organization of Animal Health, 2019*), there was a developing need for prion research and surveillance in Ethiopia and other regions in Africa, the Middle East, and the United Kingdom (*Breedlove, 2020*; *Faye, 2019*; *Gallardo & Delgado, 2021*; *Horigan et al., 2020*; *World Organization of Animal Health, 2019*; *Teferedegn, Tesfaye & Ün, 2019*). Given the increased level of local camel consumption in northern Africa, exportation of meat and milk on a world–wide scale, and lack of regulations in animal husbandry (*Teferedegn, Tesfaye & Ün, 2019*), it is crucial to develop methods for genotypic characterization of the *PRNP* gene in camels.

Previous genetic studies of dromedary camels in Algeria, Egypt, and Ethiopia (*Cherifi et al., 2017*; *Legesse et al., 2018*) reported a lack of morphological, genetic variation, and population structure indicating homogeneity in the nuclear genome of *C. dromedarius*. In addition, low variability of camel *PRNP* sequences has been reported compared to other sequences representative of dromedary camels (*Abdel-Aziem et al., 2019*; *Babelhadj et al., 2018*; *Kaluz, Kaluzova & Flint, 1997*; *Tahmoorespur & Jelokhani Niaraki, 2014*; *Xu et al., 2012*; *Zoubeyda et al., 2020*). Given the broad distribution of camel breeds across northern Africa and the apparent lack of genetic variation among breeds, it is hypothesized that Ethiopian dromedary camels will have similar *PRNP* genotypes to other dromedary camels. Therefore, the goal of this study is to determine the genotypic characterization of the *PRNP* gene in camels to ascertain the significance of predicting potential susceptibility or resistance to CPD.

## MATERIALS & METHODS

### Sampling

This project used archived DNA samples and followed protocols approved by the Texas Tech University Animal Care and Use Committee (protocol #17023-02). Tissue samples were collected as reported in (*Legesse et al., 2018*); specifically, a two cm$^2$ ear clip was obtained from domestic camels representative of *C. dromedarius*, heat denatured, stored in lysis buffer (*Longmire, Baker & Maltbie, 1997*), and deposited into a liquid nitrogen storage system for perpetuity at the Natural Science Research Laboratory (NSRL), Museum of Texas Tech University. Using *a priori* knowledge from genetic and morphological datasets (see Supplemental Information, *Legesse et al., 2018*), 50 individuals representing eight breeds of *C. dromedarius* (Afar, Aydin, Borena, Hoor, Issa, Jijiga, Kerreyu, and Liben) and 19 localities located in three Ethiopian states (Somali, Afar, and Oromia) were selected for analyses. Attempts were made to include unrelated individuals >8 years of age (*Babelhadj et al., 2018*), equal representation of sexes ($n = 20$ males, $n = 30$ females), and eight genetically and morphologically divergent breeds based on the findings in (*Legesse et al., 2018*). Additional *PRNP* sequences of *C. bactrianus* ($n = 33$), *C. dromedarius* ($n = 12$), and *C. ferus* ($n = 1$) were obtained from NCBI GenBank and were included to increase taxonomic breadth, geographic sampling, and serve as reference samples (see Supplemental Information).

### DNA sequencing

Genomic DNA was extracted from 0.1 g ear clip or 180 µl lysed ear tissue using the Qiagen DNeasy blood and tissue extraction kit (Qiagen, Hilden, Germany). The entire *PRNP* gene (768 bp) was amplified using the polymerase chain reaction (PCR) method (*Saiki et al., 1988*) with forward (5′–GCTGACACCCTCTTTATTTTGCAG–3′) and reverse (5′–GATTAAGAAGATAATGAAAACAGGAAG–3′) primers (*Kaluz, Kaluzova & Flint, 1997*), following the HotStarTaq (Qiagen Inc., Hilden, Germany) protocol. PCR reactions contained 3 µL of gDNA, 12.5 µL HotStarTaq premix, 8.3 µL of double–distilled water (ddH$_2$O), and 0.6 µL of each 10 µM primer. The thermal profile for PCR was as follows: hot start at 80 °C, initial denaturation at 95 °C for 2 min, followed by 34 cycles of denaturation at 95 °C for 30 s, annealing at 47.5–48.5 °C for 45 s, and extension at 73 °C for 1 min, with a final extension at 73 °C for 15 min.

PCR products were purified with ExoSAP–IT PCR Product Cleanup (Applied Biosystems, Foster City, CA, USA). Cycle sequencing reactions were conducted following the BigDye Terminator v3.1 protocol (Applied Biosystems, Foster City, CA, USA): 1 µL BigDye Terminator, 1 µL BigDye Buffer, 1 µL ddH$_2$O, 4 µL purified PCR product, and 3 µL of each 1 µM of primer (*Kaluz, Kaluzova & Flint, 1997*). Cycle sequencing products were purified using Sephadex filtration columns (Cytiva, Marlborough, MA, USA) and centrifugation methods, followed by dehydration. Purified sequencing products were analyzed on an ABI 3730xl automated sequencer (Eurofins Genomics LLC, Louisville, KY, USA). Resulting sequences were proofed using Sequencher 4.10.1 software (Gene Codes Corporation, Ann Arbor, MI, USA) and chromatograms generated from raw sequence reads were visually inspected to verify all nucleotide calls and identify heterozygous nucleotide

base positions. All DNA sequences obtained in this study were deposited in NCBI GenBank (OP414498–OP414547).

## Data analyses
### Characterization of PRNP

The program MEGA11 (*Tamura, Stecher & Kumar, 2021*) was used to translate the nucleotide sequences to protein, allowing for the detection of any non–synonymous substitutions. A parsimony analysis using PAUP* Version 4.0a169 (*Swofford, 2003*) was then conducted on inferred amino acids to identify synapomorphies indicative of phylogenetically informative nucleotide or amino acid replacements in the *PRNP* gene of *Camelus*.

### Genetic divergence

Genetic distance values for selected taxa were estimated for the *PRNP* dataset using the program MEGA11 (*Tamura, Stecher & Kumar, 2021*) and the Kimura 2–parameter model of evolution (*Kimura, 1980*). The resulting values were used to compare levels of genetic divergence in the *PRNP* dataset to select members of Orders Perrisodactyla and Cetartiodactyla.

### Selection on camel prion protein

DNA sequence divergence for relative contribution of neutral, negative, or positive selection was assessed using the CodeML program of PAML4.9j and PAML–X1.3.1 (*Xu & Yang, 2013*; *Yang, 2007*). Selection analyses generating dN/dS ratios ($\omega$, omega) were calculated from the codon alignments with models: M0, null; M1, neutral selection; M2, positive selection; M7, neutral selection model with seven site classes; and M8, positive selection model with eight site classes.

### Phylogenetic analyses on the PRNP gene

Given the phylogenies obtained from supertree analyses (*Bininda-Emonds et al., 2007*; *Upham, Esselstyn & Jetz, 2019*) and a mammalian speciation gene (*Roberts et al., 2022*; *Roberts et al., 2023*), the common vampire bat (*Desmodus rotundus*) was designated as the outgroup species. Representative individuals ($n = 302$) from the Order Perissodactyla (*i.e., Ceratotherium*, *Equus*, *etc.*) and Cetartiodactyla (*i.e.,* members of Suborders Mysticeti, Odontoceti, Suiformes, Tylopoda, and Ruminantia) served as ingroup taxa (see Supplemental Information). Ninety–six *Camelus* individuals (50 sampled herein and 46 acquired from GenBank, Supplemental Information) were used to assign individuals to a clade.

Eighty–eight maximum likelihood (ML) models were evaluated using jModelTest-–2.1.10 (*Darriba et al., 2012*; *Guindon & Gascuel, 2003*). The Akaike information criterion with a correction for finite sample sizes (AICc, *Burnham & Anderson, 2004*; *Hurvich & Tsai, 1989* identified the Kimura 2–parameter model of evolution (*Kimura, 1980*) plus gamma distribution model of nucleotide substitutions (K80+ $\Gamma$, –lnL = 7,546.6384) as the most appropriate for the *PRNP* dataset. A likelihood analysis was performed using RAxML Version 8.2.12 (*Stamatakis, 2014*) and the following parameters: base frequencies

($A = 0.2439$, $C = 0.2611$, $G = 0.2916$, and $T = 0.2034$), and the GTR+I+$\Gamma$ (general time reversible plus proportion of invariable sites plus gamma distribution model of nucleotide substitution). Nodal support was evaluated using the bootstrap method with 1,000 iterations (*Felsenstein, 1985*), with bootstrap values (BS) $\geq$ 65 used to indicate moderate to strong nodal support.

A ML analysis under a Bayesian inference (BI) model using MrBayes v3.2.6 (*Ronquist et al., 2012*) was conducted to generate posterior probability values (PPV). The GTR+I+$\Gamma$ nucleotide substitution model and the following parameters were used: two independent runs with four Markov–chains (one cold and three heated; MCMCMC), 10 million generations, and sample frequency of every 1,000 generations from the last nine million generated. A visual inspection of likelihood scores resulted in the first 1,000,000 trees being discarded (10% burn–in) and a consensus tree (50% majority rule) constructed from the remaining trees. PPV $\geq$ 0.95 were used to designate nodal support (*Huelsenbeck et al., 2002*).

### Phylogenetic analyses on the PRNP gene including human and resistant taxa

A BI analysis was utilized to visualize both nucleotide and amino acid topologies of *PRNP*. The armadillo (*Dasypus novemcinctus*) was designated as the outgroup species based on phylogenies obtained from supertree analyses (*Bininda-Emonds et al., 2007*; *Upham, Esselstyn & Jetz, 2019*) and a speciation gene (*Roberts et al., 2022*; *Roberts et al., 2023*). Additional taxa included representatives of the Orders Carnivora ($n = 48$), Order Chiroptera ($n = 20$), Order Eulipotyphla ($n = 6$), Order Lagomorpha ($n = 1$), Order Pholidota ($n = 1$), and Primates ($n = 1$) as well as taxa from the previous phylogenetic analyses, totaling a dataset with 578 sequences.

The GTR+I+$\Gamma$ nucleotide substitution model and the following parameters were used: two independent runs with four Markov–chains (one cold and three heated; MCMCMC), 10 million generations, and sample frequency of every 1,000 generations from the last nine million generated. A visual inspection of likelihood scores resulted in the first 1,000,000 trees being discarded (10% burn–in) and a consensus tree (50% majority rule) constructed from the remaining trees. PPV $\geq$ 0.95 were used to designate nodal support (*Huelsenbeck et al., 2002*).

# RESULTS

## Characterization of PRNP

There was one nonapeptide (PQGGGGWGQ) in the N terminal amino acid residue sites followed by four octapeptide (PHGGGWGQ) repeats beginning at codon 54 and ending at codon 94 (spans 41 amino acids). Sequence data indicated that the *PRNP* gene was monomorphic in all Ethiopian Dromedary camels. Of five nonsynonymous nucleotide substitutions, three were phylogenetically information between *C. dromedarius* and *C. bactrianus* (Table 1). Parsimony analyses indicated that these two species of camels possessed three synonymous nucleotide substitutions (T231C, T243A, and T264C) that were phylogenetically informative. Two of which (T231C and C246A) were synapomorphic

**Table 1  Differences in nucleotide and amino acid substitutions of *PRNP* between *Camelus dromedarius* and *C. bactrianus*.**

| *PRNP* nucleotide position | Nucleotide change | Amino acid |
|---|---|---|
| 231[*] | T to C | G77G |
| 243[*] | A to T | G81G |
| 246[*] | C to A | G82G |
| 264 | T to C | H88H |
| 765 | A to T | G255G |

Notes.
[*]The asterisk indicates phylogenetically informative characters.

uniting individuals assigned to *C. bactrianus*; whereas one (T243A) was synapomorphic uniting individuals assigned to *C. dromedarius*.

## Genetic divergence

Estimation of Kimura–2 parameter (*Kimura, 1980*) genetic distances (Tables 2–3), obtained from the *PRNP* dataset, indicated that the average genetic distance in *PRNP* sequences between *C. dromedarius* and *C. bactrianus* was 0.59%; whereas genetic distances within *C. dromedarius* and *C. bactrianus* were 0.03% and 0.10%, respectively (Table 2). Genetic divergences between orders (Table 2) ranged from 8.16% (Order Cetacea to Order Artiodactyla) to 11.01% (Order Perissodactyla to Order Artiodactyla). Genetic divergences within orders (Table 3) ranged from 1.06% (Order Perissodactyla) to 5.79% (Order Artiodactyla). Genetic divergences among families (Table 2) ranged from 2.31% (Family Moschidae to Family Bovidae) to 11.60% (Family Equiidae to Family Bovidae). Genetic divergences within families (Table 3) ranged from 0.42% (Family Camelidae) to 2.51% (Family Bovidae).

## Selection on camel prion protein

All models, including M0, M1, M2, M7, and M8 (*Yang, 2007*), detected pervasive, negative selection. The M0 model (null selection model) observed an omega of 0.05605. The M1 model containing two site classes ($K = 2$) detected an omega of 0.05604 at 99.99% of sites and an omega of 1.00 at 0.00001% of sites. The M2 model with three site classes ($K = 3$) detected an omega of 0.05605 at 100% of sites and two other classes with an omega of 1.0000 at 0.00% of sites. The M7 model with five site classes ($K = 5$) detected the following omegas: (1) omega = 0.03026 at 20.0% of sites, (2) omega of 0.04315 at 20.0% of sites, (3) omega of 0.05392 at 20.0% of sites, (4) omega of 0.06625 at 20.0% of sites, and (5) omega of 0.08684 at 20.0% of sites. The M8 model with five site classes ($K = 6$) detected the following omegas: (1) omega of 0.03025 at 20.0% of sites, (2) omega of 0.04315 of 20.0% of sites, (3) omega of 0.05391 at 20.0% of sites, (4) omega of 0.06624 at 20.0% of sites, (5) omega of 0.08683 at 20.0% of sites, and (6) omega of 1.00000 at 0.00001%.

## Phylogenetic analyses of the PRNP gene

Phylogenetic trees were constructed to investigate the evolutionary history of the *PRNP* gene so that signals reflecting susceptibility or resistance could be identified. Bayesian

Wright et al. (2024), *PeerJ*, DOI 10.7717/peerj.17552

**Table 2   Average genetic distances (%), using the Kimura-2 parameter (*Kimura, 1980*), among all selected taxa based on the BI and 2 ML phylogenies.** The bold value highlights the genetic distance between *Camelus bactrianus* and *C. dromedarius*.

| | 1 | 2 | 3 | 4 | 5 | 6 | 7 | 8 | 9 | 10 | 11 | 12 | 13 | 14 | 15 | 16 | 17 | 18 | 19 |
|---|---|---|---|---|---|---|---|---|---|---|---|---|---|---|---|---|---|---|---|
| 1. Chiroptera | | | | | | | | | | | | | | | | | | | |
| 2. Suidae | 12.40 | | | | | | | | | | | | | | | | | | |
| 3. Caprini, Hippotragini, Alcelaphinae | 12.48 | 11.39 | | | | | | | | | | | | | | | | | |
| 4. *Camelus dromedarius* | 10.59 | 8.56 | 9.96 | | | | | | | | | | | | | | | | |
| 5. *Camelus bactrianus* | 10.81 | 8.59 | 10.13 | **0.59** | | | | | | | | | | | | | | | |
| 6. Odontoceti | 11.36 | 9.57 | 7.83 | 8.43 | 8.60 | | | | | | | | | | | | | | |
| 7. Mystoceti | 11.22 | 9.59 | 7.90 | 8.23 | 8.35 | 3.06 | | | | | | | | | | | | | |
| 8. *Equus* | 9.59 | 9.91 | 11.42 | 9.89 | 9.97 | 10.65 | 11.38 | | | | | | | | | | | | |
| 9. *Diceros, Ceratotherium* | 9.30 | 9.82 | 11.34 | 8.59 | 8.72 | 10.09 | 10.29 | 4.46 | | | | | | | | | | | |
| 10. *Hexaprotodon, Hippotamus* | 13.25 | 10.95 | 10.45 | 8.94 | 9.08 | 7.77 | 7.58 | 11.59 | 10.55 | | | | | | | | | | |
| 11. *Lama, Vicugna* | 11.50 | 9.98 | 11.59 | 2.08 | 2.66 | 9.74 | 9.56 | 11.30 | 10.08 | 10.13 | | | | | | | | | |
| 12. Bovini | 12.68 | 11.85 | 3.18 | 10.22 | 10.39 | 8.35 | 8.51 | 12.06 | 11.54 | 10.46 | 11.76 | | | | | | | | |
| 13. Odocoileini, Cervini, Muntiacini | 12.26 | 11.17 | 2.86 | 9.34 | 9.48 | 7.95 | 7.63 | 11.55 | 11.33 | 9.89 | 10.85 | 3.87 | | | | | | | |
| 14. *Antilocapra americana* | 10.77 | 9.09 | 3.32 | 7.37 | 7.46 | 6.46 | 6.93 | 8.80 | 9.01 | 9.11 | 8.63 | 3.12 | 3.49 | | | | | | |
| 15. Tragelaphini | 12.87 | 11.19 | 3.15 | 9.75 | 9.94 | 7.84 | 8.22 | 11.18 | 10.87 | 10.15 | 11.24 | 2.97 | 3.80 | 3.26 | | | | | |
| 16. Reduncini, Antilopini | 12.03 | 10.70 | 2.76 | 9.11 | 9.27 | 7.44 | 7.93 | 10.66 | 10.18 | 10.38 | 10.43 | 2.78 | 3.57 | 1.90 | 3.03 | | | | |
| 17. Reduncini | 12.46 | 11.26 | 2.66 | 9.82 | 10.01 | 7.78 | 8.08 | 11.16 | 10.34 | 10.37 | 11.27 | 3.28 | 3.47 | 3.05 | 3.14 | 2.14 | | | |
| 18. *Moschus chrysogaster* | 12.16 | 11.53 | 2.31 | 9.36 | 9.83 | 7.47 | 7.29 | 10.77 | 11.10 | 9.32 | 10.99 | 3.28 | 2.82 | 3.54 | 3.41 | 3.17 | 3.00 | | |
| 19. Antilopini | 11.30 | 10.89 | 3.04 | 9.02 | 9.21 | 7.59 | 7.95 | 10.67 | 10.02 | 10.08 | 10.42 | 3.45 | 3.86 | 2.78 | 3.39 | 1.77 | 2.94 | 3.31 | |
| 20. *Giraffa camelopardalis* | 11.82 | 9.76 | 4.06 | 9.04 | 9.11 | 7.97 | 8.04 | 9.98 | 10.21 | 10.50 | 10.35 | 4.29 | 3.88 | 3.24 | 3.54 | 4.22 | 4.33 | 3.77 | 4.57 |

**Table 3  Average genetic distances (%), using the Kimura-2 parameter (*Kimura, 1980*), among all selected taxa based on the BI and ML phylogenies.**  The bold value highlights the genetic distance between *Camelus bactrianus* and *C. dromedarius* respectively.

| Within | GD % |
|---|---|
| Family Suidae | 1.50 |
| Caprini, Hippotragini, Alcelaphinae | 0.84 |
| *Camelus dromedarius* | **0.03** |
| *Camelus bactrianus* | **0.10** |
| Odontoceti | 1.80 |
| Mystoceti | 0.73 |
| *Equus* | 0.68 |
| *Diceros, Ceratotherium* | 1.22 |
| Hexaprotodon, Hippotamus | 0.90 |
| *Lama, Vicugna* | 2.31 |
| Bovini | 1.45 |
| Odocoileini, Cervini, Muntiacini | 1.12 |
| *Antilocapra americana* | NA |
| Tragelaphini | 1.44 |
| Reduncini and Antilopini | 1.16 |
| Reduncini | 1.32 |
| *Moschus chrysogaster* | NA |
| Antilopini | 0.53 |
| *Giraffa camelopardalis* | NA |
| Family Bovidae | 2.51 |
| Family Camelidae | 0.42 |

**Notes.**
NA, not available.

Inference (BI) and Maximum Likelihood (ML) analyses produced similar topologies; therefore, only the BI topology was depicted (Fig. 1) with posterior probability values (PPV) ≥ 0.95 and bootstrap values (BS) ≥ 65 superimposed onto the BI topology (Fig. 1). The topology of the phylogenetic tree obtained from the BI analysis indicated nodal support (PPV ≥ 0.95) for 25 of 27 nodes (Fig. 1); whereas the topology of the phylogenetic tree indicated nodal support (BS ≥ 65) for 24 of 27 nodes generated in the ML analyses (Fig. 1). Members of *Camelus* were sister to a supported clade comprised of individuals representing *Lama glama* and *Vicugna pacos*, which all are members of Camelidae. The clade representing individuals of *C. bactrianus* was supported by BI and ML analyses; however, the grouping of individuals representing *C. dromedarius* was unsupported by BI and ML analyses. The Order Perissodactyla and Order Cetartiodactyla, the Suborders Suiformes, Tylopoda, Mysticeti, and Ruminantia, and Family Antilocapridae were supported by BI and ML analyses. Within the Suborder Ruminantia, the placement of Bovidae, Cervidae, Giraffidae, and Moschidae was unresolved and may be due to limited taxon representation. The terminal nodes for the clades of Tribes: (1) Caprini, Hippotragini, and Alcelaphini, (2) Bovini, (3) Reduncini, (4) Antilopini (individuals representative of *Procapra*), and (5) Odocoileini, Cervini, and Muntiacini were supported by both BI and ML analyses. Tribes

Tragelaphini and Reduncini and Antilopini (contained individuals of *Kob* and *Gazella*) were supported by the BI analyses, but not the ML analyses at the terminal nodes. In addition, the Suborder Odontoceti was not supported by the BI analysis but was supported by the ML analysis.

### Phylogenetic analyses of the PRNP gene including human and resistant taxa

The amino acid topology from the BI analysis lacked PPV support and was not informative; therefore, this analysis was not included for discussion. The nucleotide topology described Eulipotyphla as paraphyletic with *Sorex* basal to all other taxa, except for the outgroup (*Dasypus novemcinctus*). The placement of Lagomorpha was unsupported between Eulipotyphla and Primates, which was attached to the supported lineage of Primates. A supported clade containing representatives of Old World bats (Chiroptera, Suborder Yinpterochiroptera) was followed by two supported clades containing New World bats (Chiroptera, Yangochiroptera) and the rest of Eulipotyphla. The grouping of Carnivora, Perrisodactyla, and Pholidota was basally unsupported; however, there was support at the terminal nodes. Perrisodactyla was basal to Carnivora and Pholidota with Carnivora as sister taxa to Pholidota. The rest of the topology was the same as the topology previously described above. Significant amino acid substitutions that were described in the literature are superimposed onto this topology. In addition, representatives from the above orders were aligned in a table format to identify any novel amino acid substitutions that may be implicated in prion disease.

## DISCUSSION

All 50 individuals representing eight breeds of Ethiopian *C. dromedarius* were monomorphic for the *PRNP* gene and were 100% identical to the nucleotide sequences reported for the CPD–positive individuals in Algeria (*Babelhadj et al., 2018*). Most studies (*Abdel-Aziem et al., 2019*; *Adeola et al., 2024*; *Babelhadj et al., 2018*; *Kaluz, Kaluzova & Flint, 1997*; *Tahmoorespur & Jelokhani Niaraki, 2014*; *Tahmoorespur & Jelokhani Niaraki, 2014*; *Xu et al., 2012*; *Zoubeyda et al., 2020*) report low variability in the *PRNP* gene among *Camelus* (0.42% reported herein). Three synonymous nucleotide substitutions (T231C, T243A, and T264C), identified as phylogenetically informative by Parsimony analyses, differentiated the two *Camelus* species. Further, one nonapeptide (PQGGGGWGQ) in the N terminal amino acid residue sites followed by four octapeptide (PHGGGWGQ) repeats beginning at codon 54 and ending at codon 94 (spans 41 amino acids) were determined, which is consistent with findings reported in *C. dromedarius* (*Kaluz, Kaluzova & Flint, 1997*) and *C. bactrianus* (*Xu et al., 2012*). *Zoubeyda et al. (2020)* identified one synonymous substitution (T191T, referenced using traditional amino acid terminology; *den Dunnen & Antonarakis, 2000*), and two nonsynonymous substitutions (G69S and G134E). Further, *Adeola et al. (2024)* observed a one base insertion of a thymine at position 35, which translated to a valine, and a nonsynonymous substitution (G255R).

Currently, it is unknown if these amino acid substitutions have the potential to confer resistance or susceptibility to CPD. In other well–studied organisms, such as domestic sheep

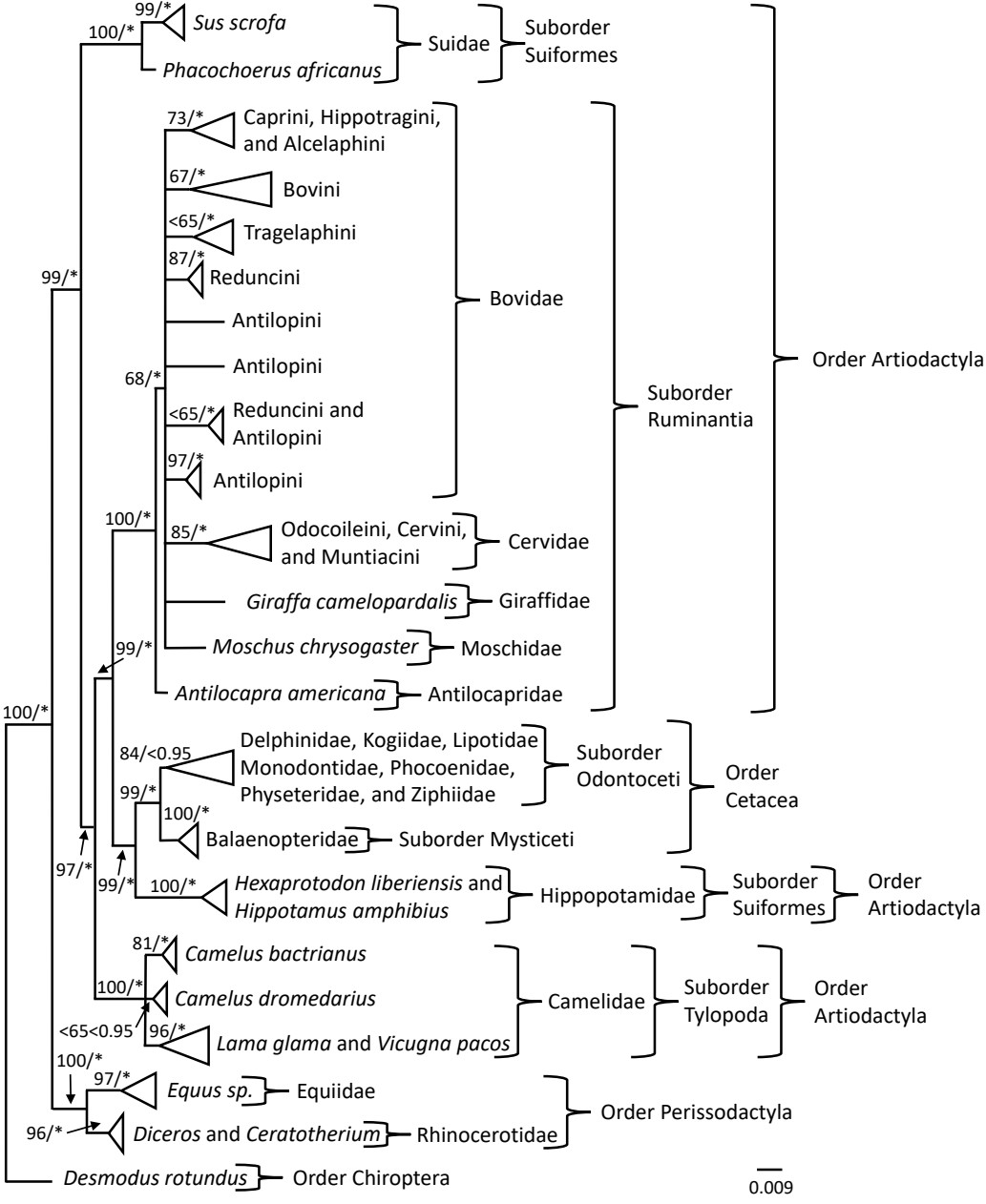

**Figure 1** *PRNP* **phylogeny of Perissodactyla and Cetartiodactyla with asterisks representing ≥0.95 Bayesian nodal support and bootstrap values ≥65 indicating moderate to strong likelihood nodal support.** Phylogeny of the exon 3 region of the prion protein gene obtained from Bayesian and Maximum Likelihood analyses of 399 sequences representing the mammalian orders of Perissodactyla and Cetartiodactyla. Bayesian posterior probability values are indicated by an * and represent ≥0.95 nodal support and likelihood bootstrap values are represented left of the slash where bootstrap values ≥75 indicate strong nodal support, bootstrap values between 65 and 75 indicate moderate nodal support, and bootstrap values ≤65 indicate low nodal support.

and goats and free–ranging and captive cervids, among others, there are known genotypes or single codons that infer resistance (R171 in sheep, V129 in humans; *Goldmann, 2008*;

*Kosami et al., 2022*; *Riek et al., 1998*; *Satoh & Nakamura, 2022*) or susceptibility (G96 and S225 in deer) to prion diseases (*Goldmann, 2008*; *Kobayashi et al., 2015*; *Orge et al., 2021*, Table 4). For example, in domestic sheep, there are five categories that range from high risk for prion infection to complete resistance pertaining to three codons (amino acids 136, 154, and 171) and their associated amino acid substitutions (*Goldmann, 2008*). However, in some cases, ARR sheep are susceptible to prions derived from BSE (*Houston et al., 2003*) and atypical scrapie (*Buschmann et al., 2004*), respectively. In free–ranging populations of North American deer (*Odocoileus*), a *PRNP* genotype associated with complete resistance has yet to be identified. However, G96S in white–tailed deer (*O. virginianus*) and S225F in mule deer (*O. hemionus*) provided potential evidence for a less efficient conversion from $PrP^C$ to $PrP^{Sc}$ and delayed onset of clinical signs in inoculation and knock–out studies in mice (Table 4, *Arifin et al., 2023*; *Jewell et al., 2005*; *Johnson et al., 2011*; *Otero et al., 2021*; *Perucchini et al., 2008*; *Roh et al., 2022*). Other amino acids may be implicated in the presentation of prion diseases. Seven codon positions, which differ among mammals known to be resistant or susceptible, are hypothesized that may contribute either resistance or susceptibility among select mammalian lineages (Table 4) and supports the hypothesis of positive selection at the amino acid level of *PRNP* (*Premzl & Gamulin, 2009*) while maintaining an overall strong purifying selection of *PRNP* across Mammalia (*Seabury et al., 2004*).

The lack of nucleotide diversity in *PRNP* of *Camelus* may be due to a relatively recent evolutionary history of CPD and therefore has not had sufficient time to produce and accumulate allelic variation and develop a potential resistance to $PrP^{Sc}$ (*Zoubeyda et al., 2020*). For example, in some domesticated livestock (*i.e.,* sheep and goats, *Goldmann, 2008*), there is considerable genetic variability (2.51% reported herein) in the *PRNP* gene where TSEs have been documented since the 1700's (*Plummer, 1946*). However, selection tests identified pervasive negative selection acting on the *PRNP* gene in *Camelus*, with similar omega values to cattle (*Slate, 2005*); thus, there was no detection of selection towards particular nucleotides or amino acids for resistance or susceptibility to CPD. Although there was consensus that purifying selection has acted on the evolution of *PRNP* in ruminants, there were cases in which some ungulates, mainly sheep and goats, possessed site-specific positive selection (*Premzl & Gamulin, 2009*; *Slate, 2005*).

Another factor that may explain this low genetic variability is that *C. bactrianus* and *C. dromedarius* naturally (and artificially in livestock operations) interbreed and produce fertile hybrid offspring characterized by hybrid vigor based on anthological and hybridization studies (*Dioli, 2020*; *Lado et al., 2019*; *Tapper, 2011*). Given that *C. bactrianus* can successfully interbreed and produce viable offspring with *C. dromedarius*, hybrid *Camelus* can contract Middle East Respiratory Syndrome (*Lau et al., 2020*), and the close genetic relationship of *C. ferus* to both *C. bactrianus* and *C. dromedarius* (*Roberts et al., 2022*), *C. bactrianus*, *C. ferus*, and hybrid *Camelus* (free–ranging or in captivity) also could be susceptible to CPD, considering only *C. dromedarius* has been detected with CPD (*Babelhadj et al., 2018*).

Based on the BI and ML phylogenetic analyses, Order Perissodactyla (*Ceratotherium*, *Diceros*, and *Equus*) and Suborder Suiformes (*Sus* and *Phacochoerus*) are the only ungulate

**Table 4  Alignment of the prion protein gene across several mammalian taxa.**

| Species | Tribe | 1↓                                                          58 |
|---|---|---|
| *Homo sapiens* [1,2,3] | | – – –MALGCWMLVLFVATWSDLGLCKKRPKP–GGWNTG–GSRYPGQGSPGGNRYPPQGG |
| *Oryctolagus cuniculus* [4,5,6,7] | | . .MAH . .Y. . .L. . . . . . . .V. . . . . . .G. . . . . . . . . . . . . .S. . . . |
| *Acinonyx jubatus* [8] | | MVKGHI.G.I. . . . . . . . . .V. . . . . . . . .G. . . . . . . . . . . . . . . . . . . . . |
| *Canis familiaris* [7,9] | | MVKSHI.G.I.L. . . . . . . .V. . . . . . . . . . . . .G. . . . . . . . . . |
| *Eptesicus fuscus* [9,10] | | MVKSLV.G.I. . . . . . . . . .V. . . . . . .G.– –. . . . . . . . . . . |
| *Pteropus alecto* [9,10] | | MVKNYI.G.I. . . . . . . . . . . . . . .V. . . . . . .G. .GSS. . . . . . . . . . |
| *Equus caballus* [11,12] | | MVKSHV.G.I. . . . . . . . . .V. . . . . . . . . . . . . . . . . . . . . . . . . . . . |
| *Ceratotherium sinum*[*] | | MVKSHV.G.I. . . . . . . . . .V. . . . . . . . . .G. . . . . . . . . . . . . . . . |
| *Vicugna pacos*[*] | | MVKSHM.S.I. . . . .V. . . .V. . . . . . . . . .G. . . . . . . . . . . . . . . . |
| *Sus scrofa* [13,14] | | MVK**S**HI.G.I. . . . .A. . .I. . . . . . . . .G. . . . . . . . . . . . . . . |
| *Phacochoerus africanus*[*] | | MVKSHI.G.I. . . . .A. . .I. . . . . . . . .G. . . . . . . . . . . . . . . . |
| *Camelus spp.* | | MVKSHM.S.I. . . . .V. . . .V. . . . . . . . . .G. . . . . . . . . . . . . . . |
| *Antilocapra americana*[*] | | MVKSHI.S.I. . . . . .M. . .V. . . . . . . . .G. . . . . . . . . . . . . . . |
| *Giraffa camelopardalis*[*] | | . . .– – – – – – – – – – – – – – – – – – – – . . . . . . .G. . . . . . . . . . . . . . |
| *Bos spp.* [13,15] | Bovini | MVKSHI.S.I. . . . . .M. . .V. . . . . . . . .G. . . . . . . . . . . . . . . . |
| *Ovis aries* [13,16] | Caprini | MVKSHI.S.I. . . . . .M. . .V. . . . . . . . .G. . . . . . . . . . . . . . . . |
| *Capra hircus* [13,16] | Caprini | MVKSHI.S.I. . . . . .M. . .V. . . . . . . . .G. . . . . . . . . . . . . . . . |
| *Kobus megaceros*[*] | Reduncini | MVKSHI.S.I. . . . . .M. . .V. . . . . . . . .G. . . . . . . . . . . . . . . . |
| *Tragelaphus angasii* [17,18] | Tragelaphini | MVKSHI.S.I. . . . . .M. . .V. . . . . . . . .G. . . . . . . . . . . . .S. . . . |
| *Procapra gutturosa*[*] | Antilopini | MVKSHI.S.I. . . . . .M. . .V. . . . . . . . .G. . . . . . . . . . . . . . . . |
| *Hippotragus niger*[*] | Hippotragini | MVKSHI.R.I. . . . . .M. . .V. . . . . . . . .G. . . . .V. . . .C. . . . . . |
| *Connochaetes taurinus*[*] | Alcelaphini | MVKSHI.S.I. . . . . .M. . .V. . . . . . . . .G. . . . . . . . . . . . . . . . |
| *Odocoileus spp.* [19,20] | Odocoileini | MVKSHI.S.I. . . . . .M. . .V. . . . . . . . .G. . . . . . . . . . . . . . . . |
| *Cervus elaphus* [21] | Cervini | MVKSHI.S.I. . . . . .L. . .M. . .V. . . . . . . . .G. . . . . . . . . . . . . . |
| *Cervus nippon* [22] | Cervini | MVKSHI.S.I. . . . . .M. . .V. . . . . . . . .G. . . . . . . . . . . . . . . . |
| *Muntiacus reevesi* [23] | Muntiacini | MVKSHI.S.I. . . . . .M. . .V. . . . . . . . .G. . . . . . . . . . . . . . . . |
| *Moschus chrysogaster*[*] | | MVKSHI.S.I. . . . . .M. . .V. . . . . . . . .G.– . . . . . . . . . . . . . .A |
| *Hippopotamus amphibius*[*] | | . . .– – – – – – – – – – – – – – – – – – – – —. . . . .G. . . . . . . . . . . . . . |
| *Balaenoptera musculus*[*] | | MVKSHIAN.I. . . . . . .C. .M. . . . . . . .G. . . . . . . . . . . . . . . |
| *Tursiops truncates*[*] | | MVKSHIAN.I. . . . . . . . .M.F. . . . . . .G. . . . . . . . . . . . . . . . |

**Table 4** (*continued*)

| Species | Tribe | 59 | 116 ↓ |
|---|---|---|---|
| *Homo sapiens* | | GGWGQP– –GGGWGQPHG– – – – – – – –GGWGQP– –GG– –GWGQPHGG– –GWGQ– –GGGTH | |
| *Oryctolagus cuniculus* | | . . . . . . . . . . . . . . . . . . . . . . . . . . . . . . . . . . . . . . . . . . . . . . . . . . . . . . | |
| *Acinonyx jubatus* | | . . . . . .HA. . . . . . . .AG. . . . . . . . . . . . . .HA. . . . . . . . . . .A. .G. . . . . . . . . . . | |
| *Canis familiaris* | | . . . . . . . . . . . . . . . . . . . . . . . . . . . . . . . . . . . . . .G. . . . . . . . . .S. | |
| *Eptesicus fuscus* | | . . . . . .QG. . . . . . . . . . . . . . . . . . . . . . . . . . . .GG. . . . . . . .GG. . . .PH. . .S. | |
| *Pteropus alecto* | | . . . . . . . . . . . . . . .GGWGQPHG. . . . . . . . . . . . . . . . . .-.G. . . . . . . . . | |
| *Equus caballus* | | . . . . . . . . . . . . . . . . . . . . . . . . . . . . . . . . . . . . . .G. . . . . .–S. | |
| *Ceratotherium sinum* | | . . . . . . . . . . . . . . . . . . . . . . . . . . . . . . . . . . . . . .G. . . . . .–S. | |
| *Vicugna pacos* | | . . . . . . . . . . . . . . .GGWGQPHG. . . . . . . . . . . . . . . .–.G. . . . . . . . . | |
| *Sus scrofa* | | . . . . . . . . . . . . . . . . . . . . . . . . . . . . . . . . . . . . . .G. . . . . . . .S. | |
| *Phacochoerus africanus* | | . . . . . . . . . . . . . . . . . . . . . . . . . . . . . . . . . . . . . .G. . . . . . . .S. | |
| *Camelus dromedarius* | | . . . . . . . . . . . . . . . . . . . . . . . . . . . . . . . . . . .–.G. . . . . . . . . .A. | |
| *Antilocapra americana* | | . . . . . . . . . . . . . . .GGWGQPHG. . . . . . . . . . . . . . . . .G. . . . . . .–. . | |
| *Giraffa camelopardalis* | | . . . . . . . . . . . . . . .GGWGQPHG. . . . . . . . . . . . . . . . .G. . . . . . .–. . | |
| *Bos indicus* | Bovini | . . . . . . . . . . . . . . .GGWGQPHG. . . . . . . . . . . . . . . . .G. . . . . . .–. . | |
| *Ovis aries* | Caprini | . . . . . . . . . . . . . . . . . . . . . . . . . . . . . . . . . . . . . .G. . . . . .–S. | |
| *Capra hircus* | Caprini | . . . . . . . . . . . . . . . . . . . . . . . . . . . . . . . . . . . . . .G. . . . . .–S. | |
| *Kobus megaceros* | Reduncini | . . . . . . . . . . . . . . . . . . . . . . . . . . . . . . . . . . . . . .G. . . . . .–. . | |
| *Tragelaphus angasii* | Tragelaphini | . . . . . . . . . . . . . . . . . . . . . . . . . . . . . . . . . . . . . .G. . . . . .–. . | |
| *Procapra gutturosa* | Antilopini | . . . . . . . . . . . . . . .GGWGQPHG. . . . . . . . . . . . . . . . .G. . . . . .–. . | |
| *Hippotragus niger* | Hippotragini | . . . . . . . . . . . . . . . . . . . . . . . . . . . . . . . . . . . . . .G. . . . . .–. . | |
| *Connochaetes taurinus* | Alcelaphini | . . . . . . . . . . . . . . . . . . . . . . . . . . . . . . . . . . . . . .G. . . . . .–. . | |
| *Odocoileus virginianus* | Odocoileini | . . . . . . . . . . . . . . . . . . . . . . . . . . . . . . . . . . . . . .G. . . . . .–. . | |
| *Cervus elaphus* | Cervini | . . . . . . . . . . . . . . . . . . . . . . . . . . . . . . . . . . . . . .G. . . . . .–. . | |
| *Cervus nippon* | Cervini | . . . . . . . . . . . . . . . . . . . . . . . . . . . . . . . . . . . . . .G. . . . . .–. . | |
| *Muntiacus reevesi* | Muntiacini | . . . . . . . . . . . . . . . . . . . . . . . . . . . . . . . . . . . . . .G. . . . . .–S. | |
| *Moschus chrysogaster* | | . . . . . . . . . . . . . . . . . . . . . . . . . . . . . . . . . . . . . .G. . . . . .–. . | |
| *Hippopotamus amphibius* | | . . . . . . . . . . . . . . . . . . . . . . . . . . . . . . . . . . . . . .G. . . . . . . . . | |
| *Balaenoptera musculus* | | . . . . . . . . . . . . . . . . . . . . . . . . . . . . . . . . . . . . . .G. . . . . . . . . | |
| *Tursiops truncatus* | | . . . . . . . . . . . . . . . . . . . . . . . . . . . . . . . . . . . . . .G. . . . . . . . . | |

| Species | Tribe | 117↓ 174 |
|---------|-------|----------|
| *Homo sapiens* | | SQWNKPSKPKTNMKHMAGAAAAGAVVGGLGGY**M**LGSAMSRPIIHFG**S**DYEDRYYRENM |
| *Oryctolagus cuniculus* | | N. .G. . . . . . .S. . .V. . . . . . . . . . . . . . . . . . . . . . . .L. . . .N. . . . . . . . . |
| *Acinonyx jubatus* | | . . .G. . . . . . . . . . . . . . . . . . . . . . . . . . . . . . **.** . .L. . . .N. . **.** . . . . . . |
| *Canis familiaris* | | **G**. .G. .N. . . . . . . .V. . . . . . . . . . . . . . . . . . . .L. . . .N. . . . . . . . . |
| *Eptesicus fuscus* | | N. . . . .N. . . . . . . . . . . . . . . . . . . . . . . . . . . . .AM. . .NE. . . . . . . . . |
| *Pteropus alecto* | | . . . . . . . . . . . .L. .V. . . . . . . . . . . . . . .N. .LL. . .N. . . . . . . . . |
| *Equus caballus* | | G**...**. . . . . . . . . .V. . . . . . . . . . . . . . . . . . .L. . . .N. . . . . . . . . |
| *Ceratotherium sinum* | | G. . . . . . . . . . . . . . . . . . . . . . . . . . . . .L. . .N. . . . . . |
| *Vicugna pacos* | | G. . . . . . . . .S. . .V. . . . . . . . . . . . . . . . .L. . . .N. . . . . . . |
| *Sus scrofa* | | G. . . . . . . . . . . . .V. . . . . . . . . . . . . . .L. . . . . . . . . . |
| *Phacochoerus africanus* | | G. . . . . . . . . . . . .V. . . . . . . . . . . .N. .L. . . . . . . . . . |
| *Camelus dromedarius* | | G. . . . . . . . .S. . .V. . . . . . . . . . . . . . . .L. . . .N. . . . . . . |
| *Antilocapra americana* | | . . . . . . . . . . . . .V. . . . . . . . . . . . . . .L. . . .N. . . . . . . . |
| *Giraffa camelopardalis* | | G. . . . . . . . . . . . . . . . . . . . . . . . . .L. . .N. . . . . . . . |
| *Bos indicus* | Bovini | G. . . . . . . . . . . .V. . . . . . . . . . . . . . .L. . . .N. . . . . . . |
| *Ovis aries* | Caprini | . . . . . . . . . . . .V. . . . . . . . . . . . . . .L. . . .N. . . . . . . |
| *Capra hircus* | Caprini | . . . . . . . . . . . .V. . . . . . . . . . . . . .L**...** .**N**. . . . . . . |
| *Kobus megaceros* | Reduncini | . . . . . . . . . . . .V. . . . . . . . . . . . . . .L. . .N. . . . . . . |
| *Tragelaphus angasii* | Tragelaphini | G. . . . . . . . . . . .V. . . . . . . . . . . . . .L. . . . . . . . . . |
| *Procapra gutturosa* | Antilopini | . . . . . . . . . . . .V. . . . . . . . . . . . . . .L. . .N. . . . . . . |
| *Hippotragus niger* | Hippotragini | . . . . . . . . . . . .V. . . . . . . . . . . . . . .L. . .N. . . . . . . |
| *Connochaetes taurinus* | Alcelaphini | . . . . . . . . . . . . . . . . . . . . . . . . . .L. . . .N. . . . . . . |
| *Odocoileus virginianus* | Odocoileini | . . . . . . . . . . . .V . . . . . . . . . . . . . .L. . . .N. . . . . . . |
| *Cervus elaphus* | Cervini | . . . . . . . . . . . .V. . . . . . . . . . . . . .L. . . .N. . . . . . . |
| *Cervus nippon* | Cervini | . . . . . . . . . . . .V. . . . . . . . . . . . . .L. . . .N. . . . . . . |
| *Muntiacus reevesi* | Muntiacini | . . . . . . . . . . . .V. . . . . . . . . . . . . .L. . . .N. . . . . . . |
| *Moschus chrysogaster* | | N. . . . . . . . . . . .V. . . . . . . . . . . . . .L. . . .N. . . . . . . |
| *Hippopotamus amphibius* | | G. . . . . . . . . . . . . . . . . . . . . . . . . .L. . . . . . . . . . |
| *Balaenoptera musculus* | | N. . . . . . . . . . . .V. . . . . . . . . . . . . .L. . . . . . . . . . |
| *Tursiops truncatus* | | N. . . . . . . . . . . .V. . . . . . . . . . . . . .L. . . . . . . . . . |

**Table 4** (*continued*)

| Species | Tribe | 175 ↓                                                                            232 |
|---|---|---|
| *Homo sapiens* | | HRYPNQVYYRPMDEHSNQNNFVHDCVNITIKQHTV–TTTTKGENFTETDVKMMERVVE |
| *Orytolagus cuniculus* | | Y. . . . . . . . . .V.QY. . . .S. . . . . . . . . . . .V. . . . . . . . . . . . . . . . . . .I.I. . . . . . |
| *Acinonyx jubatus* | | Y. . . . . . . . . .V.QY. . . . . . . . . . . . . . .VR. . . . . . . . . . . . . . . . . . .M.I. . . . . . |
| *Canis familiaris* | | Y. . . **E** . . . . . .V.QY. . . . . .R. . . . . .V. . . . . . . . . . . . . . . . . . .M.I. . . . . . |
| *Eptesicus fuscus* | | N.F.**E**R. . .K.V.QYN. . . . . .R. . . . . .V. . . . . . . . . . . .D. . . . . . . .I. . . . . . |
| *Pteropus alecto* | | Y. . . . . . . . . .V.QYN. . .S. . . . . . . . . . .V. . . . . . . . . . . . . . . . . . . . . . . . . . . |
| *Equus caballus* | | Y. . . . . . . . . .V**S**QY. . . . . . . . . . . . . . .V. . . . . . . . . . . . . . . . . . .I. . . . . . |
| *Ceratotherium sinum* | | Y. . . . . . . . . .VS.Y. . . . . . . . . . . . .V. . . . . . . . . . . . . . . . . . . . . . . . . . . |
| *Vicugna pacos* | | Y. . . . . . .K.VS.Y. . . .S. . . . . . . . . .V. . . . . . . . . . . . . . . . . . . . . . . . . . . |
| *Sus scrofa* | | Y. . . . . . . . . .V.QY. . . .**S**. . . . . . . . . .V. . . . . . . . . . . . . . . . . . .I. . . . . |
| *Phacochoerus africanus* | | Y. . . . . . . . . .V.QY. . . .S. . . . . . . . . .V. . . . . . . . . . . . . . . . . . .I. . . . . |
| *Camelus dromedarius* | | Y. . . . . . .K.V.QY. . . .S. . . . . . . . . .V. . . . . . . . . . . . . . . . . . . . . . . . . . |
| *Antilocapra americana* | | Y. . . . . . . . . .V.QY. . . .T. . . . . . . . . .V. . . . . . . . . . . . . . . . . . .I. . . . . . |
| *Giraffa camelopardalis* | | Y. . . . . . . . . .V.QY. . . .T. . . . . . . . . .V. . . . . . . . . . . . . . . . . . .I. . . . . . |
| *Bos indicus* | Bovini | . . . . . . . . . .V.QY. . . . . . . . . . . . .V.E. . . . . . . . . . . . . . . . . . .I. . . . . . |
| *Ovis aries* | Caprini | Y. . . . . . . . . .V.**Q**Y. . . . . . . . . . . . .V. . . . . . . . . . . . . . . . . . .I.I. . . . . . |
| *Capra hircus* | Caprini | Y. . . . . . . . . .V.QY. . . . . . . . . . . . .V. . . . . . . . . . . . . . . . . . .I.I. . . . . . |
| *Kobus megaceros* | Reduncini | Y. . . . . . . . . .V.QY. . . . . . . . . . . . .V. . . . . . . . . . . . . . . . . . .I.I. . . . . . |
| *Tragelaphus angasii* | Tragelaphini | Y. . . . . . . . . .V.QY. . . . . . . . . . . . .V. . . . . . . . . . . . . . . . . . .I. . . . . . |
| *Procapra gutturosa* | Antilopini | Y. . . . . . . . . .V.QY. . . . . . . . . . . . .V. . . . . . . . . . . . . . . . . . .I. . . . . . |
| *Hippotragus niger* | Hippotragini | Y. . . . . . . . . .V.QY. . . . . . . . . . . . .V. . . . . . . . . . . . . . . . . . .I.I. . . . . . |
| *Connochaetes taurinus* | Alcelaphini | Y. . . . . . . . . .V.QY. . . . . . . . . . . . .V. . . . . . . . . . . . . . . . . . .I.I. . . . . . |
| *Odocoileus virginianus* | Odocoileini | Y. . . . . . . . . .V.QYN. . .T. . . . . . . . . .V. . . . . . . . . . . . . . . . . . .I. . . . . . |
| *Cervus elaphus* | Cervini | Y. . . . . . . . . .V.QYN. . .T. . . . . . . . . .V. . . . . . . . . . . . . . . . . . .I. . . . . . |
| *Cervus nippon* | Cervini | Y. . . . . . . . . .V.QYN. . .T. . . . . . . .**V**. . . . . . . . . . . . . . . . . . .I. . . . . . |
| *Muntiacus reevesi* | Muntiacini | Y. . . . . . . . . .V.QYN. . .T. . . . . . . . . .V. . . . . . . . . . . . . . . . . . .I. . . . . . |
| *Moschus chrysogaster* | | Y. . . . . . . . . .V.QY. . . . . . . . . . . . .V. . . . . . . . . . . . . . . . . . .I.I. . . . . . |
| *Hippopotamus amphibius* | | . . . . . . . . . .V.QYH. . . . . . . . . . . . .V. . . . . . . . . . . . . . . . .I. .I. . . . . |
| *Balaenoptera musculus* | | Y. . .S. . . . . .V.QYNS. .S. . . . . . . . . .V. . . . .T. . . . . . . . . . . . . .I. . . . . . |
| *Tursiops truncatus* | | Y. . . . . . . . . .V.QYN. . .S. . . . . . . . . .V. . . . .T. . . . . . . . . . . . . .I. . . . . . |

| Species | Tribe | 233 ↓ | 274 |
|---------|-------|-------|-----|
| *Homo sapiens* | | QMCITQY**E**RESQAYYQRGSSMVLFSSPPVILLISFLIFLIVG | |
| *Oryctolagus cuniculus* | | . . . . . . .QQ. . . .A. . .AAGVL. . . . . . . . . . . . . . | |
| *Acinonyx jubatus* | | . . .V. . .QK. .E. . . . . .A.AI. . .P. . . . . .L.L. .L. .G. | |
| *Canis familiaris* | | . . .V. . .QK. .E. . . . . .A.AI. . .P. . . . . . . .L. .L. . . . | |
| *Eptesicus fuscus* | | E. .T. . .QK.Y. .A. . . .A.VI. . . . . . . . . . . . . . . . . | |
| *Pteropus alecto* | | . . . . . . .QQ. .R.A.H. .A.V.V. . . . . . . . . . . . . . . | |
| *Equus caballus* | | . . . . . . .QK.YE.FQ. . .A.V. . . . . . . . . .V. . . . . . . . . | |
| *Ceratotherium sinum* | | . . . . . . .Q. .YE.FQ. . .A.V. . . . . . . . . .V. . . . . . . . . | |
| *Vicugna pacos* | | . . . . . . .Q. .Y. .S.G. .A.V. . . . . . . . . . . . . . . . . | |
| *Sus scrofa* | | . . . . . .QK.YE. .A. . .A.VI. . . . . . . . . . . . . .L. . . . . | |
| *Phacochoerus africanus* | | . . . . . .QK.YE. .A. . .A.VI. . . . . . . . . . . . . . . . . . | |
| *Camelus dromedarius* | | . . . . . . .Q. .Y. .S.G. .A.V. . . . . . . . . . . . . . . . . | |
| *Antilocapra americana* | | . . . . . . .Q. . . . . . . . . . .A.VI. . . . . . . . . . . . . . | |
| *Giraffa camelopardalis* | | . . . . . . .Q. . . .E. . . . . .A.VI. . . . . . . . . . .– – – – – | |
| *Bos spp.* | Bovini | . . . . . . .Q. . . . . . . . . .A.VI. . . . . . . . . . . . . . | |
| *Ovis aries* | Caprini | . . . . . . .Q. . . . . . . . . .A.VI. . . . . . . . . . . . . . | |
| *Capra hircus* | Caprini | . . . . . . .**Q**. . . . . . . . . .A.VI. . . .P. . . . . . . . . . . . . | |
| *Kobus megaceros* | Reduncini | . . . . . . .Q. . . .E. . . . . .A.VI. . . . . . . . .F. . . . . . . . | |
| *Tragelaphus angasii* | Tragelaphini | . . . . . . .Q. . . .E. . . . . .A.VI. . . . . . . . . . . . . . | |
| *Procapra gutturosa* | Antilopini | . . . . . . .Q. . . . . . . . . .A.VI. . . . . . . . . . . . . . | |
| *Hippotragus niger* | Hippotragini | . . . . . . .Q. . . . . . . . . .A.VI. . . . . . . . . . . . . . | |
| *Connochaetes taurinus* | Alcelaphini | . . . . . . .Q. . . . . . . . . .A.VI. . . . . . . . . . . . . . | |
| *Odocoileus spp.* | Odocoileini | . . . . . . .Q. . . . . . . . . .A.VI. . . . . . . . . . . . . . | |
| *Cervus elaphus* | Cervini | . . . . . . .Q. . . . . . . . .A.VI. . . . . . . . . . .L. . . . . . . | |
| *Cervus nippon* | Cervini | . . . . . . .Q. . . . . . . . . .A.VI. . . . . . . . . . . . . . | |
| *Muntiacus reevesi* | Muntiacini | . . . . . . .Q. . . . . . . . . .A.VI. . . . . . . . . . . . . . | |
| *Moschus chrysogaster* | | . . . . . . .Q. . . . . . . . . .A.VI. . . . . . . . . . . . . . | |
| *Hippopotamus amphibius* | | . . . . . . .QK.Y. . . .E. .A.V.V. . . .L.V. . . .–– – – – – – – | |
| *Balaenoptera musculus* | | . . . . . . .Q. .Y. . . . . . .A.VI. . . . . . . . . . . . . . . | |
| *Tursiops truncatus* | | . . . . . . .Q. .YE. . . . . .A.VI. . . . . . . . . . . . . . . | |

**Table 4** (*continued*)

**Notes.**

*Unknown risk of prion disease in certain taxa are indicated with an asterisk.

Highlighted in bold are amino acids that are implicated in resistance or susceptibility to prion disease. Arrows indicate potential codons that may infer resistance or susceptibility to prion diseases.

[1]*Kosami et al. (2022)*
[2]*Riek et al. (1998)*
[3]*Satoh & Nakamura (2022)*
[4]*Chianini et al. (2012)*
[5]*Fernandez-Borges et al. (2012)*
[6]*Nisbet et al. (2010)*
[7]*Myers, Cembran & Fernandez-Funez (2020)*
[8]*Kim et al. (2020a)*
[9]*Fernandez-Borges, Erana & Castilla (2018)*
[10]*Stewart et al. (2012)*
[11]*Kim & Jeong (2018)*
[12]*Kim et al. (2020b)*
[13]*Ibeagha-Awemu et al. (2008)*
[14]*Meng et al. (2005)*
[15]*Seabury et al. (2004)*
[16]*Goldmann (2008)*
[17]*Kirkwood & Cunningham (1994)*
[18]*Cunningham et al. (2004)*
[19]*Jewell et al. (2005)*
[20]*Otero et al. (2021)*
[21]*Perucchini et al. (2008)*
[22]*Roh et al. (2022)*
[23]*Nalls et al. (2013)*

lineages basal to the Suborder Tylopoda (*Camelus*, *Lama*, and *Vicugna*). The Camelidae lineage is weakly supported and forms a polytomy among the three species available for these analyses. Additionally, the clade of *C. bactrianus* was supported and differed from *C. dromedarius* by three phylogenetically informative nucleotide positions. Further, the Order Cetartiodactyla was strongly supported for cetacean and artiodactylid lineages. Although the phylogeny depicted herein is a gene tree, the *PRNP* gene indicated that Hippopotamidae is sister to cetaceans, not the Suborder Suiformes, which is supported by previous studies using morphological, mitochondrial genomes, supertrees, and divergence dating analyses (*Boisserie, Lihoreau & Brunet, 2005*; *Hassanin et al., 2012*; *Price, Bininda-Emonds & Gittleman, 2005*; *Zurano et al., 2019*). This phylogeny tracks the evolutionary history of prion disease among ungulate taxa and CPD appears to be on its own evolutionary trajectory basal to BSE, scrapie, and CWD. The most recent detection of prion disease in camels (*Babelhadj et al., 2018*) may be indicative of the general susceptibility to prion disease in individuals representing Cetartiodactyla.

Based on the *PRNP* phylogeny, Equiidae (and Rhinocerotidae) and Suidae are the only ungulate taxa that are more basal than *Camelus*. However, these two groups have shown to be highly resistant to prion diseases (*Espinosa et al., 2021*; *Kim & Jeong, 2018*; *Myers, Cembran & Fernandez-Funez, 2020*). Could it be that the susceptibility of prion diseases in ungulates began with camels or a common ancestor of members of Cetartiodactyla (to the exclusion of Suidae)? Other resistant mammals include rabbits and some canids (*Myers, Cembran & Fernandez-Funez, 2020*). Although the hypothesis of truly resistant mammals to prion diseases is contentious in the literature (*Chianini et al., 2012*; *Fernandez-Borges et al., 2012*; *Nisbet et al., 2010*), mammals may be susceptible to prion diseases with *in vivo*

challenges under highly favorable laboratory conditions, but never encounter or develop prion disease in nature (*Myers, Cembran & Fernandez-Funez, 2020*). Alignment of the prion protein (PrP) indicates some amino acid substitutions that may be informative for the potential and onset of disease (Table 4). For example, some canids and some chiropterans (only insectivorous bats in Vespertilionidae; EA Wright, 2022, unpublished data) both possess either aspartic acid (D) or glutamic acid (E) at codon 179 (aligned to human PrP). Although the proposed resistance in Chiroptera has been hypothesized (*Fernandez-Borges, Erana & Castilla, 2018*; *Stewart et al., 2012*), genotypic and *in vivo* challenges need to be explored in further studies to determine the breadth of the N179D/E among taxa representative of Chiroptera and the potential resistance to prion diseases.

Phylogenetic methodologies, in conjunction with documented genetic disease profiles, have the potential to predict taxonomic clusters of resistance or potential susceptibility to diseases (*Cajimat et al., 2007*; *Cajimat et al., 2011*; *Fulhorst et al., 2002*). Based on this premise, manifestation of prion disease in wild populations and agricultural operations may only affect members of Cetartiodactyla (Fig. 1), specifically excluding pigs (*Espinosa et al., 2021*; *Fernandez-Borges et al., 2012*; *Myers, Cembran & Fernandez-Funez, 2020*) and their relatives (*i.e., Sus* and *Phacochoerus*). The distribution of phylogenetically informative characters of *PRNP* with clinical and reported cases of prion disease suggest that the clades containing camels, cetaceans, hippos, cervids, bovids, and others should be susceptible to prion diseases (Fig. 1).

## CONCLUSIONS

Although more studies are needed to determine if there is a difference in the genotypic profile of *PRNP* among CPD–positive and CPD–negative individuals, this preliminary study demonstrates the genetic similarity in the *PRNP* gene between Algerian and Ethiopian camels. The lack of nucleotide and corresponding amino acid differentiation, as well as a lack of genetic diversity between Ethiopian and Algerian dromedaries leads us to conclude that dromedaries from both of these regions have equivalent susceptibility to developing *PRNP* mutations, infection, and transmission rates. Considering *Camelus* products, such as milk and meat, are distributed widely in Africa and Europe (*World Organization of Animal Health, 2019*), CPD transmission may mirror the BSE outbreak, which was the causal agent of variant CJD (*Mead et al., 2009*), if health and safety precautions are ignored. Camelid antibodies have been used in trials for the treatment of neurodegenerative diseases and others (*David, Jones & Tayebi, 2014*; *Jones et al., 2010*; *Tayebi et al., 2010*) and possess unique characteristics that allow these antibodies to cross the blood–brain barrier (*Hamers-Casterman et al., 1993*; *Steeland, Vandenbroucke & Libert, 2016*). Further, knock–out and inoculation trials with mice using *PRNP* of *Camelus* need to be conducted to examine the susceptibility of *Camelus* to BSE, CWD, and other prion diseases and the zoonotic potential for transmission from camels to other artiodactylids as well as humans (*Watson et al., 2021*).

Considering the novelty of CPD, surveillance studies should be implemented in regions where abattoirs are common. Collaborations among international universities,

federal agencies, and agricultural workers are essential to these research areas. Further investigations in Ethiopia are in stages of development to: (i) identify potential routes of CPD transmission and document standard practices involved in camel husbandry, butchery, and sale, (ii) incorporate a human dimensions aspect, with current CPD awareness among abattoir meat inspectors, (iii) assess the prevalence of CPD, and (iv) generate policy recommendations for CPD. The implementation of a CPD surveillance program will contribute to the overall One Health of humans, camels, and the environment.

## ACKNOWLEDGEMENTS

Thanks to the TTU High Performance Computing Center for providing bioinformatics support and access to software and hardware. Thanks to H. Garner and K. MacDonald of the NSRL for providing tissues used in this project. Thanks to C. D. Dunn and M. R. Mauldin for generating the cytochrome–*b* and microsatellite data. Thanks to Jigjiga University for their assistance in collecting tissues for the project.

### Funding

This work was funded by the Bobby Baker Memorial Scholarship for Excellence in Scientific and Genomics Research to Madison B. Reddock and a State of Texas line item (Biological Database) to Robert D. Bradley. The funders had no role in study design, data collection and analysis, decision to publish, or preparation of the manuscript.

### Grant Disclosures

The following grant information was disclosed by the authors:
Bobby Baker Memorial Scholarship for Excellence in Scientific and Genomics Research.
State of Texas line item (Biological Database).

### Competing Interests

The authors declare there are no competing interests.

### Author Contributions

- Emily A. Wright conceived and designed the experiments, performed the experiments, analyzed the data, prepared figures and/or tables, authored or reviewed drafts of the article, and approved the final draft.
- Madison B. Reddock performed the experiments, prepared figures and/or tables, authored or reviewed drafts of the article, and approved the final draft.
- Emma K. Roberts performed the experiments, analyzed the data, prepared figures and/or tables, authored or reviewed drafts of the article, and approved the final draft.
- Yoseph W. Legesse conceived and designed the experiments, prepared figures and/or tables, authored or reviewed drafts of the article, and approved the final draft.
- Gad Perry conceived and designed the experiments, prepared figures and/or tables, authored or reviewed drafts of the article, and approved the final draft.

- Robert D. Bradley conceived and designed the experiments, analyzed the data, prepared figures and/or tables, authored or reviewed drafts of the article, and approved the final draft.

### Animal Ethics

The following information was supplied relating to ethical approvals (i.e., approving body and any reference numbers):

Texas Tech University Animal Care and Use Committee

### Data Availability

The sequences are available at NCBI GenBank: OP414498–OP414547.

### Supplemental Information

Supplemental information for this article can be found online at http://dx.doi.org/10.7717/peerj.17552#supplemental-information.

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
