# Peer review of "Genetic characterization of the prion protein gene in camels (Camelus) with comments on the evolutionary history of prion disease in Cetartiodactyla"

_PeerJ, doi:10.7717/peerj.17552_

## Round 0.1 · original submission · Major Revisions

Please address the issues pointed out by the reviewers and amend the manuscript accordingly.

Reviewer 1 ·

Basic reporting

Perhaps the final tables are too heavy and large for me and could have been better in supplementary material.

Experimental design

Introduction: More needs to be added about PRNP gene polymorphisms found in different species and how they are associated with disease in sheep, goats, deer and even horses as recent studies of the latter have emerged.
Material and methods:
It would remain to be added whether the samples obtained have any relationship between them or are totally independent.

Validity of the findings

No comments

Additional comments

This is a very interesting research article, based on a genetic study of African camelids, to detect possible variations in the PRNP gene, making them more susceptible or resistant to prion diseases.

Reviewer 2 ·

Basic reporting

.

Experimental design

.

Validity of the findings

.

Additional comments

In this study, the authors have examined the PRNP gene in 50 individuals representing eight breeds of C. dromedarius in Ethiopian states.

There are some critical points that the authors may consider:
1. The PRNP genes in 65 camels from Nigeria were published in the journal by Yahyaoui MH last month.
Adeola AC et al. Single nucleotide polymorphisms (SNPs) in the open reading frame (ORF) of prion protein gene (PRNP) in Nigerian livestock species. BMC Genomics. 2024 Feb 14;25(1):177.
2. Yahyaoui MH groups have already performed phylogenetic analysis in their paper.
3. So, This manuscript has no novelty.

Reviewer 3 ·

Basic reporting

see below

Experimental design

see below

Validity of the findings

The results do not offer any insight into the evolutionary history of prion disease in Cetartiodactyla. The authors should reserve their commentary until they possess data. The experimental design is incapable of supporting the statement. Undoubtedly there is selection on the prion protein, but it is unlikely that the selection arises from prion diseases. This may be “affirming the consequent.”

The camel prion protein does not seem unique or unusual in any manner. Rather, the variation within it is encapsulated across the other species and seems typical based upon its phylogenetic placement.

Broadly, all mammals that express the prion protein are susceptible to prion diseases. Conversely, Mammals that do not express the prion protein are resistant. The hypothesis that a particular prion protein primary structure confers broad disease resistance is incompatible with the data and fails to account for prion strain variants. For example, the authors discuss sheep scrapie, “in domestic sheep, there are five categories that range from high risk for prion infection to complete resistance pertaining to three codons (amino acids 136, 154, and 171).” This statement is minimally incomplete and wrong with respect to complete resistance. ARR sheep are less susceptible to the prion strains described as “classical scrapie” and are susceptible to BSE prions and atypical scrapie (12774113, 15019257). Similarly in white-tailed deer, the G96S polymorphism affects the progression of typical CWD strains, and yet is sensitive other CWD strains (32111742).

The authors state “Ethiopian camels may be highly susceptible to CPD under confinement, such as pastoral environments.” Are camels with passports from other countries also highly susceptible to CPD under confinement? Perhaps Ethiopian Camels are not susceptible when free ranging?

---

## Round 0.2 · Major Revisions

Please carefully address remaining concerns of the reviewer #3 and amend your manuscript accordingly.

Reviewer 1 ·

Basic reporting

No comment

Experimental design

No comment

Validity of the findings

No comment

Additional comments

No comment

Reviewer 3 ·

Basic reporting

no comment

Experimental design

no comment

Validity of the findings

Perhaps unsurprisingly, I am unconvinced by the Author's responses. There is no doubt that the prion protein is conserved. Whether this is due to prion disease and more specifically, prion diseases in camels seems dubious. I suppose it is a fine suggestion to make. I do not find evidence in the paper to support the broader claim. I suppose the authors are free to make whatever claim they want from the 50 sequences and the associative study design. For my part, I am far more open to other explanations such as spurious factors in design or sampling, the effects of domestication, positive selection of a linked gene involved in reproduction or a multitude of other explanations, so few having been ruled out.

I think it would be important for the authors to describe the free ranging camel populations in Ethiopia and the extent of their selection that is free from the influence of human domestication. How many camels in Ethiopia are not confined? Is there any "natural" reproduction and selection?

"It is thought that prion diseases spread more rapidly in confined settings, such as slaughterhouses for camels ..." --Prion diseases take years to manifest. How can disease be spread in a slaughterhouse? ".. or deer breeding facilities in the US." -- there is no shortage of CWD in wild cervids in N. America.

Certainly, there are species for which a prion disease has not yet been documented. Does this mean that those species are resistant or that they are resistant to the particular prions agents against which they have been challenged? I am not sure that one can reliably make the jump from, dogs were resistant to the prions with which they were challenged, to the broader claim that "dogs are highly resistant to prion diseases." Perhaps, dogs have not been challenged with many prions owing to their unique status as companion animals and restrictions on their use as experimental animals.

---

## Round 0.3 · Minor Revisions

Please address the remaining concerns of the reviewer and amend the manuscript accordingly.

Reviewer 3 ·

Basic reporting

na

Experimental design

na

Validity of the findings

na

Additional comments

The authors may want to reference some of the work done to understand the normal function of the prion protein. A robust literature exists describing prion proteins and their cellular and neurological functions e.g. PMID:35163156.

The manuscript is improved by the discussion of the domestication of camels and the acknowledgement that there are not wild camels in Africa. This idea that CPD is spreading in abattoirs is unjustified. How long does a camel survive in an abattoir (maybe a week?) Their response to this is nonsensical. The authors built a straw man concerning a higher probability of disease contact and transmission of CPD in closed settings and then use data from other prion diseases to drive a conclusion. Again, what camels are free ranging? Where is the data showing this in CPD?

With respect to the concept of highly resistant species, the authors rightfully state that the issue is contentious. The contentiousness arises because resistance/susceptibility is a complex idea that demands rigorous experimentation. It is not a simple binary determination. There are data available for resistance, but it is not as simple as finding the primary sequence of the prion protein gene and then making claims. The authors are not the first to align the prion proteins of cherry-picked species, call some of them susceptible and some of them resistant, highlight amino acid substitutions, and make wild claims regarding their potential for disease. Resistance to which prion strain? The putative resistance of dogs is a great example as the claim is consistent with the lack of observed prion disease in dogs, and as the authors claim tgMice. Still, these arguments are not dispositive. The experimental design of a tgMouse transmission experiment cannot conclude that the host species is resistant when no transmission is observed. A multitude of evidence demonstrates this with tgHuman mouse models of CJD. Classically, when tgHuman mice were first constructed (PMID:7937921), they were "resistant" to human prions, "Tg(HuPrP) Mice Are Resistant to Human Prions." The failure of known prion strains to transmit to mice with synonymous prion proteins has been repeatedly observed. When we see transmission in a tgMouse, we can conclude something, but when we see nothing, we cannot draw firm conclusions.

---

## Round 0.4 · accepted · Accept

All remaining issues were adequately addressed, and the revised manuscript is acceptable now.